# Water Quality Evaluation and Prediction Based on a Combined Model

Guimei Jiao [1],*, Shaokang Chen [1], Fei Wang [1], Zhaoyang Wang [2], Fanjuan Wang [1], Hao Li [1], Fangjie Zhang [1], Jiali Cai [1] and Jing Jin [1]

1 School of Mathematics and Statistics, Lanzhou University, Lanzhou 730000, China
2 College of Earth and Environmental Sciences, Lanzhou University, Lanzhou 730000, China
* Correspondence: gmjiao@lzu.edu.cn

**Abstract:** Along with increasingly serious water pollution, water environmental problems have become major factors that hinder the sustainable development of our economy and society. Reliable evaluation of water quality and accurate prediction of water pollution indicators are the key links in water resource management and water pollution control. In this paper, the water quality data of Lanzhou Xincheng Bridge section in the Yellow River Basin and Sichuan Panzhihua Longdong section in the Yangtze River Basin were used to establish a water quality evaluation model and a prediction model. For the water quality evaluation model, we constructed the research samples by means of equal intervals and uniform distribution of interpolated water quality index data according to *Environmental Quality Standards for Surface Water*. The training samples were determined by a stratified sampling method, and the water quality evaluation model was established using a T-S fuzzy neural network. The experimental results show that the highest accuracy achieved by the evaluation model in water quality classification was 94.12%. With respect to the water quality prediction model, we propose ARIMA-WNN, which combines the autoregressive integrated moving average model (ARIMA) and a wavelet neural network (WNN) with the bat algorithm (BA) to determine the optimal weight of each individual model. The experimental results show that the highest prediction accuracy of ARIMA-WNN is 68.06% higher than that of the original model.

**Keywords:** water quality evaluation; water quality forecasting; T-S fuzzy neural network; combined model

## 1. Introduction

Water is the foundation of human existence and the driving force for social stability and a nation's prosperity. However, water resource management has been ignored and forgotten for a long time. It was not until the mid-19th century that, due to the rapid development of industry, water pollution became increasingly serious and water resource management became increasingly prominent. [1]. Since then, the declining water quality of rivers, lakes and groundwater has become a global problem. Although an increasing number of countries has begun to attach importance to water resources and implement a series of protection measures for the sustainable development of water resources, the water resources environment is still deteriorating, with increasing pollution and waste caused by economic development, the acceleration of urbanization and population growth.

The river pollution situation is serious. The water quality level has fallen to IV or worse in 31.4% of the more than 208,000 km of managed river sections in China and below class V in 14.9% of managed sections, indicating that water resources have completely lost their potential for daily use [2]. Of the ten major river basins in China, only some in the southwest and northwest have moderate water quality (categories I to III), and the major river systems in the north, such as the Yellow River, Liao River and Huai River, are rated IV or V. The declining self-purification ability of rivers and deteriorating industrial wastewater

management have further worsened the water quality of small tributaries flowing into the major rivers of our country.

Lakes are also heavily polluted. The water quality of nearly half of the 62 key lakes in the country is of grades IV and V or inferior. The three major lakes in China, Taihu Lake, Chaohu Lake and Dianchi Lake, are polluted to varying degrees, in states of mild, moderate and severe pollution, respectively, with total phosphorus and chemical oxygen demand representing the main pollutants.

The groundwater quality situation is also worrying. In China's major cities, 27 percent of centralized drinking water sources do not meet official standards. Among the 5118 groundwater monitoring points in various provinces and cities across the country, the proportion of poor and extremely poor water quality is more than half, threatening people's daily water use [3,4].

Water environmental problems have become a major factor hindering the sustainable development of China's economy and society, and the effective treatment of water pollution and the rational management of water resources are urgent problems to be solved. The accurate prediction of water quality indicators and the reliable evaluation of water quality grades are the basis for understanding the current water quality status and taking corresponding protection measures, so water quality prediction and evaluation have great practical significance.

In this paper, we take the water quality of Lanzhou Xincheng Bridge section in Yellow River Basin and Longdong section in Yangtze River Basin as the research object, establish a water quality evaluation model and propose a new water quality prediction model.

A T-S fuzzy neural network was used to establish the evaluation model combined with the relevant water quality information of the two basins. In the process of model training, an innovative method of interpolating water quality index data with equal intervals and uniform distribution was adopted to construct research samples, and the method of stratified sampling was used to construct training samples. The trained model was applied to water quality evaluation of Lanzhou Xincheng Bridge section in the Yellow River Basin and Longdong section in the Yangtze River Basin. A total of 52 groups were randomly selected from the real water quality index data from 2004 to 2015, and the results of water quality status were output and compared with the real water quality grade to prove the effectiveness and generalizability of the evaluation model.

Furthermore, a new model is proposed for water quality prediction, which combines the autoregressive integrated moving average (ARIMA) model and the wavelet neural network (WNN) mode with the bat algorithm to determine the optimal weight of each individual model. The combined model was used to predict the water quality indices of Lanzhou Xincheng Bridge section in the Yellow River Basin and Longdong section in the Yangtze River Basin. First, 624 weekly monitoring data points of each indicator from 2004 to 2015 were used as the training set, and 52 data points from 2016 were used as the validation set. ARIMA and WNN were used for prediction. The empirical mode decomposition (EMD) algorithm was used to denoise the data before WNN prediction [5]. Secondly, the bat algorithm was used to determine the optimal weight; the final prediction result was the weighted sum of the prediction results of ARIMA and WNN. Then, the prediction results of the combined model were compared and analyzed relative to the prediction results of three individual models (backpropagation (BP), neural network and least squares support vector machine (LSSVM)) to prove the ability of the proposed combined model for water quality prediction [6,7]. Finally, the prediction results of each index in 2016 were substituted into the previously established water quality evaluation model, and the output results had a high coincidence rate with the real water quality grade, which further verified the effectiveness of the evaluation model.

## 2. Literature Review

### 2.1. Research Status of Water Quality Evaluation

At present, multivariate statistical methods are widely used in water quality evaluation and analysis abroad. Singh K P et al. (2005) applied multivariate statistical methods to the water quality evaluation of eight monitoring sites of the Gomti River in India from 1999 to 2001, demonstrating the advantages of multivariate statistical methods in processing and evaluating a large number of complex water quality datasets and obtaining effective water quality evaluation results [8]. Shrestha S et al. (2007) collected a total of 14,976 data points measuring 12 water quality indicators at 13 different monitoring points from 1995 to 2002 using multivariate statistical tools for spatiotemporal variable analysis of a large set of complex water quality data of the Fuji River. The water quality status of the 13 observation points was divided into three categories by stratified cluster analysis: mild, moderate and severe pollution [9]. Zhang X et al. (2011) used the monthly data of 23 indicators from 16 different monitoring points in southwest Kowloon, Hong Kong, from 2000 to 2007, employing hierarchical cluster analysis to divide the 12 months into two periods and the 16 monitoring points into three categories. Discriminant analysis provides analysis results from both spatial and temporal aspects. Among the 23 indicators, 4 are the main factors affecting the temporal distribution, and 8 indicators are the main factors affecting the spatial distribution [10]. Ogwueleka T C (2015) used principal component analysis, cluster analysis and factor analysis to study the water quality of the Kaduna River and analyze the potential pollution factors of the river [11]. The multivariate statistical method is a classical method for water quality assessment and management, but it cannot provide comprehensive information about water quality. On this basis, in this study, the selected neural network constantly updates and iterates the model parameters, adjusts the weights and thresholds to the state that can output the optimal results and applies the trained model to the evaluation task.

### 2.2. Research Status of Water Quality Prediction

Previously proposed water quality prediction models are based on qualitative analysis. Water quality prediction was first studied in 1925, when Phelps and Streeter proposed the S-P model to track BOD-DO changes in water quality. Since then, with the worsening global water pollution and drinking water crises, an increasing number of water quality prediction models have been proposed. However, due to the complexity of water environments, it is difficult to obtain accurate prediction results with traditional mathematical models, so scholars began to use neural networks to predict water quality. Singh KP et al. (2009) used the monthly data of 11 water quality indicators from 8 different monitoring points over 10 years to establish two BP neural network models, 11-23-1 and 11-11-1, to calculate the levels of dissolved oxygen and biochemical oxygen demand of Gomti River in India and indirectly judge the water quality [12]. Seo IW et al. (2016) used an artificial neural network model to predict eight water quality indicators downstream of Cheongpyeong Dam [13]. Zhang ran et al. (2013) established a GM(1,1) model to predict the water quality of the Yellow River estuary from 2012 to 2015 [14]. Zhang Ying et al. (2015) took the section of Shanghai Qingpu Urgent Water Port in Taihu Lake Basin as an example and applied the gray model after residual correction of an extreme learning machine regression model for the prediction of water quality indicators. They used the data of the first 100 days of 2013 of six water quality indicators, including dissolved oxygen and chemical oxygen demand, to predict the data of the 101st to 110th days [15]. Xu Hongmin et al. (2007) proposed a weighted support vector regression model to predict the concentration of permanganate in Taihu Lake Basin using the same method; the results showed that the prediction accuracy of this algorithm was higher than that of SVM and RBF neural network alone [16]. According to the idea of neural network weighting introduced in the abovementioned literature, in this study, we designed and implemented a mixed model to predict water quality.

## 3. Materials and Methods

### 3.1. T-S Fuzzy Neural Network

T-S fuzzy neural network is a new fuzzy neural network proposed by Takagi and Sugeno in 1985 [17]. Fuzzy reasoning rules are adopted in '*if then*' form:

$$R^i : If\ x_1\ is\ A_1^i,\ x_2\ is\ A_2^i, \dots,\ x_k\ is\ A_k^i\ then\ y_i = p_0^i + p_1^i x_1 + \dots + p_k^i x_k \tag{1}$$

where '*if then*' are the front part and back part of fuzzy rules, respectively (the former is the input part of fuzzy rules, and the latter is the determined output); $A_j^i$ is a fuzzy set of fuzzy models; $p_j^i (j = 1, 2, \dots, k)$ is a fuzzy model parameter; and $y_i$ is the output of the fuzzy rules. The process shows that the output is a linear combination of inputs, as shown in Figure 1.

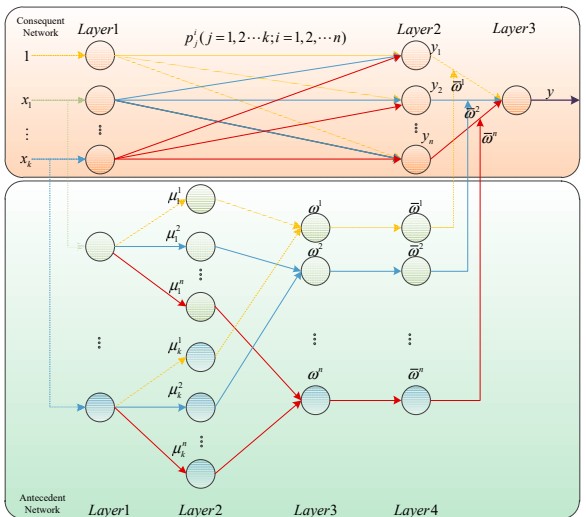

**Figure 1.** Structural diagram of T-S fuzzy neural network.

### 3.2. ARIMA

The autoregressive integrated moving average (ARIMA) model was proposed by Box and Jenkins in the 1970s [18]. It is based on the autoregressive model (AR) proposed by Yule in 1927 and the combination of the moving average model (MA) and autoregressive moving average model (ARMA) with AR and MA proposed by Walker in 1931 [19–21]. In this model, the future value of a variable is considered a linear combination of past values and past errors:

$$y_t = \theta_0 + \varphi_1 y_{t-1} + \varphi_2 y_{t-2} + \dots + \varphi_p y_{t-p} + \varepsilon_t - \theta_1 \varepsilon_{t-1} - \theta_2 \varepsilon_{t-2} - \dots - \theta_q \varepsilon_{t-q} \tag{2}$$

$$\varphi(B) \nabla^d y_t = \theta(B) \varepsilon_t \tag{3}$$

where $y_t$ is the actual value; $\varepsilon_t$ is the random error at time $t$; $\varphi_i$ and $\theta_j$ are the coefficients; $p$ and $q$ are the orders of autoregressive and sliding average polynomials, respectively; $B$ represents the lag operator, where $\Delta_d = (1 - B)^d$; $d$ is the number of differences; and $\varphi(B)$ and $\theta(B)$ are defined as:

$$\varphi(B) = 1 - \varphi_1 B - \varphi_2 B^2 - \dots - \varphi_p B^p \tag{4}$$

$$\theta(B) = 1 - \theta_1 B - \theta_2 B^2 - \dots - \theta_q B^q \tag{5}$$

### 3.3. Wavelet Neural Network

Wavelet neural network (WNN) is a neural network model that combines wavelet transform and artificial neural network, which replaces the excitation function of the

traditional neural network with the wavelet basis function. It was first proposed in 1992 by Zhang Q and Benveniste A of LRISA, a famous French information science institute [22]. The combination of wavelet transform and neural network provides unique advantages. In recent years, it has been widely used in nonlinear function approximation, dynamic modeling and non-stationary time series prediction. $X_1, X_2, \ldots, X_k$ are input parameters, $\omega_{ij}$ and $\omega_{jk}$ are connection weights and $Y_1, Y_2, \ldots, Y_m$ are predicted outputs. The formula for calculating the output of the hidden layer is:

$$h(j) = h_j \left[ \frac{\sum_{i=1}^{k} \omega_{ij} x_i - b_j}{a_j} \right], j = 1, 2, \ldots, l \tag{6}$$

where $h_j$ is the wavelet basis function; $a_j$ and $b_j$ are the scale factor and translation factor of the wavelet basis function, respectively; $\omega_{ij}$ is the connection weight between the input layer and the hidden layer; and $h(j)$ is the output value of the node of the seventh hidden layer. The calculation formula of the output layer is:

$$y(k) = \sum_{j=1}^{l} \omega_{jk} h(j) \quad k = 1, 2, \ldots, m \tag{7}$$

where $\omega_{jk}$ is the connection weight between the hidden layer and the output layer, and $y(k)$ is the output value.

### 3.4. ARIMA-WNN

Actual time series data often have both linear and nonlinear characteristics, and ARIMA or WNN alone cannot reflect the dual linear and nonlinear characteristics of time series. In order to simultaneously utilize the good linear fitting ability of the differential autoregressive moving average model and the powerful nonlinear relationship mapping ability of the wavelet neural network model, we combine ARIMA and WNN methods.

Assuming $y_t (t = 1, 2, \ldots, L)$ is the actual value of the time series, L is the number of sample points, and $\hat{y}_t$ and $\hat{y}_{it} (i = 1, 2, \ldots, N, t = 1, 2, \ldots, L)$ is the predicted value of the combined model and the first single method, respectively; then:

$$\hat{y}_t = \sum_{i=1}^{N} \lambda_i \hat{y}_{it} \tag{8}$$

where $\lambda_i$ is the weight of the prediction method, and $\sum_{i=1}^{N} \lambda_i = 1$. The weight coefficients of each of the component models in the combined model are determined by solving the following optimization problems:

$$Min \sum_{t=1}^{L} (y_t - \hat{y}_t)^2, \quad s.t. \sum_{i=1}^{N} \lambda_i = 1, \ 0 \leq \lambda_i \leq 1 \tag{9}$$

On this basis, we propose a new combination model composed of the ARIMA model and WNN [23]. The optimal weight of a single model is obtained by the bat algorithm, and the predicted value of the combination model is expressed as follows:

$$P_{\Sigma} = \lambda_1 P_{WNN} + \lambda_2 P_{ARIMA} \tag{10}$$

where $P_{\Sigma}$ is the final predicted value; $\lambda_1, \lambda_2, P_{WNN}, P_{ARIMA}$ are the weight coefficients and predicted values of WNN and ARIMA models, respectively; and $\lambda_1 + \lambda_2 = 1$, as shown in Figure 2.

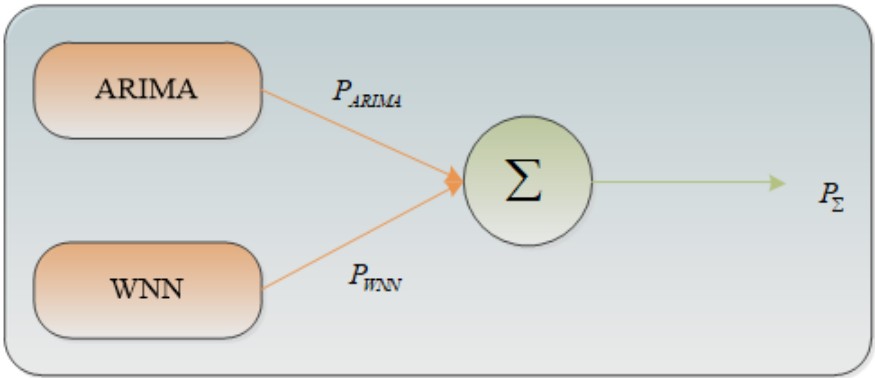

**Figure 2.** ARIMA—WNN structure.

*3.5. Bat Algorithm*

The bat algorithm (BA) is a swarm intelligence optimization algorithm that simulates the behavior of bats to hunt down prey, which was proposed by Yang X S in 2010 [24]. Let the bat search for prey in the $n$ dimensional space at the moment of $t - 1$; the flight speed and position of bat $i$ are $v_i^{t-1}$, $x_i^{t-1}$, respectively; then, the update rules of the bat's flight speed ($v_i^t$) and location at time $t$ are:

$$f_i = f_{min} + (f_{max} - f_{min}) * \beta \tag{11}$$

$$v_i^t = v_i^{t-1} + \left( x_i^{t-1} - x^* \right) * f_i \tag{12}$$

$$x_i^t = x_i^{t-1} + v_i^t \tag{13}$$

Among them, $\beta \in [0, 1]$ is a uniformly distributed random number; $f_{max}$ and $f_{min}$ are the maximum and minimum search frequency, respectively; the bat's pulse search frequency is $f_i$; and $f_i \in [f_{min}, f_{max}]$. The bat algorithm controls the prey hunting range of bats by adjusting $f_i$ and controls the global search in the whole updating process, which is the optimal solution for the current bat population.

For a local search, the bat algorithm is completed by random disturbance. Each bat randomly selects one solution from the current optimal solution set as the current optimal solution ($x_{old}$); then, we use the following formula to update the position to obtain a new solution:

$$x_{new} = x_{old} + \varepsilon A^t \tag{14}$$

where $\varepsilon \in [-1, 1]$ is a random number, and $A^t$ is the average loudness of all bats at time $t$.

*3.6. Materials*

(1)   ADF

The ADF test determines whether the series is stationary by checking whether the sum of the autoregressive coefficients is 1 [25]. Compared with the DF test, which can only be used to determine whether the AR(1) model is stationary, the ADF test can be used to determine the stationarity of the AR(P) model. The hypothesis test is established as:

$$H_0 : \rho = 0$$

$$H_1 : \rho < 0$$

The test statistic is:

$$\tau = \frac{\hat{\rho}}{S(\hat{\rho})} \tag{15}$$

where $\rho = \varphi_1 + \Lambda + \varphi_p - 1$, and $S(\hat{\rho})$ is the sample standard deviation of parameter $\rho$.

(2)  AF

The autocorrelation function describes the correlation between the random sequence at time $t$ and the value at time $t - k$ [26].

$$\rho_k = \frac{Cov(y_t, y_{t-k})}{\sqrt{Var(y_t)}\sqrt{Var(y_{t-k})}} = \frac{E[(y_t-\mu)(y_{t-k}-\mu)]}{\sqrt{Var(y_t)}\sqrt{Var(y_{t-k})}} = \frac{\gamma_k}{\sigma^2} \tag{16}$$

where $E(y_t) = \mu, \sigma^2 = E(y_t - \mu)^2$, and $\gamma_k$ denotes the covariance of $y_t$ and $y_{t-k}$.

(3)  PAF

The partial autocorrelation function shows that for sequence $y_t$ in determining $y_{t-1}$, $y_{t-2}, \ldots, y_{t-k+1}$, the correlation between $y_t$ and $y_{t-k}$ is denoted by $\varphi_{kk}$ [26]:

$$\varphi_{kk} = \begin{cases} \gamma_1 & k = 1 \\ \frac{\gamma_k - \sum_{j=1}^{k-1} \varphi_{k-1,j}\gamma_{k-j}}{1 - \sum_{j=1}^{k-1} \varphi_{k-1,j}\gamma_j} & k = 2, 3, \ldots \end{cases} \tag{17}$$

(4)  The AIC criterion

The AIC criterion was proposed by Japanese statistician Hiroji Akike as a measure of model fit excellence [27].

$$AIC = n \log \sigma^2 + 2(p + q) \tag{18}$$

where $n$ is the number of samples; $\sigma^2$ is the sum of squares of the fitted residuals; and $p$ and $q$ are the orders of the AR and MA models, respectively. Models are established from low-order to high-order according to the ARIMA values, and the AIC value of each model is calculated. According to the criterion, the model with the lowest AIC value is the optimal model.

(5)  MAE

Mean absolute error (MAE) is a measure of accuracy for regression [28]. It sums up absolute values of errors and divides them by the total number of values. It gives equal weight to each error value. The formula for calculating MAE is shown in Equation (19).

$$MAE = \frac{\sum (|\hat{y}_i - y_i|)}{n} \tag{19}$$

(6)  MAPE

The reason why mean absolute percentage error can describe accuracy is that it is often used as a statistical index to measure the accuracy of prediction [29].

$$MAPE = \frac{1}{n} \sum \left( \left| \frac{\hat{y}_i - y_i}{y_i} \right| \right) * 100\% \tag{20}$$

(7)  RMSE

Root mean squared error (RMSE) is the square root of MSE and scales the values of MSE to the ranges of observed values [28]. It is estimated according to Equation (21).

$$RMSE = \sqrt{\frac{\sum (|\hat{y}_i - y_i|)^2}{n}} \tag{21}$$

(8)  AI

In this paper, a new index (AI, accuracy improvement) is introduced to show the improvement of prediction ability of the combined model compared with the single model [30]. $S_i$ and $S_c$ are the MAPE of the single model and combined model, respectively.

$$RI = \frac{|S_i - S_c|}{S_i} \times 100\% \tag{22}$$

## 4. Results and Discussion

### 4.1. Water Quality Evaluation Model

#### 4.1.1. Data Preprocessing

The three water quality monitoring indicators used in this paper are from the data center of the Ministry of Environmental Protection (http://datacenter.mep.gov.cn/index (accessed on 1 September 2021)): dissolved oxygen (DO), chemical oxygen demand permanganate ($COD_{Mn}$) and ammonia nitrogen ($NH_3$-N). The corresponding water quality grades of each index value are shown in Table 1:

**Table 1.** Environmental quality standards for surface water (GB3838-2002).

| Classification | Class I | Class II | Class III | Class IV | Class V | Class Inferior V |
|---|---|---|---|---|---|---|
| $DO/(mg \cdot L^{-1}) \geq$ | 7.5 | 6.0 | 5.0 | 3.0 | 2.0 | rest |
| $COD_{Mn}/(mg \cdot L^{-1}) \leq$ | 2.0 | 4.0 | 6.0 | 10 | 15 | rest |
| $NH_3$-N$/(mg \cdot L^{-1}) \leq$ | 0.15 | 0.50 | 1.0 | 1.5 | 2.0 | rest |

In order to solve the problem of inadequate sample size due to only taking the water quality evaluation grading standard as the research sample, the water quality index data are interpolated by the method of equal interval and uniform distribution [31]. For the convenience of modeling, the output value is continuous, and its value range is (0.5,6.5). The relationship between output value and water quality grade is shown in Table 2.

**Table 2.** Corresponding water quality grade of output values.

| Output Value | Water Quality Grade |
|---|---|
| $0.5 < y \leq 1.5$ | Class I |
| $1.5 < y \leq 2.5$ | Class II |
| $2.5 < y \leq 3.5$ | Class III |
| $3.5 < y \leq 4.5$ | Class IV |
| $4.5 < y \leq 5.5$ | Class V |
| $5.5 < y \leq 6.5$ | Class inferior V |

#### 4.1.2. Model Building Process

In this study, T-S fuzzy neural network is used to evaluate water quality. The number of input and output nodes of the model is determined by the input and output dimensions of training samples. According to the index data considered in this paper, the input and output dimensions are determined to be three and one, respectively, so the number of input and output nodes is three and one, respectively. Through trial-and-error method, the number of hidden layer nodes is determined to be six, so the structure of water quality evaluation model is as follows: 3–6–1 [32]. The four coefficients ($P_0$, $P_1$, $P_2$ and $P_3$), the width (b) and center (c) of the membership function were randomly initialized.

After normalizing the input data, training of each parameter in the model was started [33]. After 100 iterations, the network finally converged, and the training error was $3.46 * 10^{-4}$.

#### 4.1.3. Water Quality Evaluation of Lanzhou Xincheng Bridge Section

The process of water quality evaluation is realized in MATLAB, and the confusion matrix and water quality evaluation chart are output, as shown in Figure 3 [34]:

$$\begin{bmatrix} 0 & 0 & 0 & 0 & 0 & 0 \\ 1 & 41 & 3 & 0 & 0 & 0 \\ 0 & 1 & 6 & 0 & 0 & 0 \\ 0 & 0 & 0 & 0 & 0 & 0 \\ 0 & 0 & 0 & 0 & 0 & 0 \\ 0 & 0 & 0 & 0 & 0 & 0 \end{bmatrix}$$

**Figure 3.** Water quality grade confusion matrix of Lanzhou Xincheng Bridge section.

In classification models, confusion matrices are often used to observe and judge the number of correct and incorrect classifications. As shown in Figure 3, among the 52 weeks of random sampling, correct in water quality evaluation was achieved for 47 weeks, and the total correct judgment rate reached 90.38%. The correct prediction rate of class II and class III water quality was 91.11% and 85.71%, respectively. Class II water quality was misjudged as class I for one week, class II water quality was misjudged class as III for three weeks, and class III water quality was misjudged class II for one week.

The total number of correct and incorrect judgments and the corresponding water quality grade can be determined according to the above analysis, but it is not clear in which week the incorrect judgment occurred. This information can be obtained from the water quality evaluation chart presented below.

In Figure 4, the *x*-axis indicates the number of weeks; the *y*-axis indicates the water quality evaluation; and the green dot and the red dots represent the real water quality grade and the water quality grade determined by the model, respectively. Overlapping of the red and green dots indicates that the water quality level is correctly judged for that week, whereas divergence indicates an incorrect judgment. As shown in Figure 4, the water quality grade is concentrated in class II and class III, which is not accidental because during the whole time period from 2004 to 2015, the water quality grade is mainly class II and class III. Five of the fifty-two water quality grades were incorrectly judged; details are shown in Table 3.

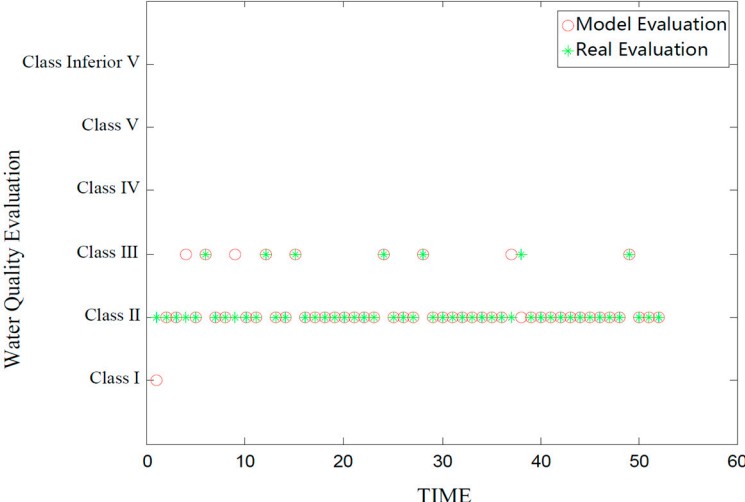

**Figure 4.** Water quality evaluation of Lanzhou Xincheng Bridge section.

**Table 3.** Water quality grade misjudgment of Lanzhou Xincheng Bridge section.

| Week | True Water Quality | Misjudged Quality |
|---|---|---|
| 1 | II | I |
| 4 | II | III |
| 9 | II | III |
| 37 | II | III |
| 38 | III | II |

### 4.1.4. Water Quality Evaluation of Longdong Section

Similarly, the trained T-S fuzzy neural network is applied to the water quality evaluation of Longdong section in Panzhihua, Sichuan, in the Yangtze River Basin, with a confusion matrix and water quality evaluation map as output, as shown in Figures 5 and 6, respectively.

$$\begin{bmatrix} 32 & 2 & 0 & 0 & 0 & 0 \\ 3 & 14 & 0 & 0 & 0 & 0 \\ 0 & 1 & 0 & 0 & 0 & 0 \\ 0 & 0 & 0 & 0 & 0 & 0 \\ 0 & 0 & 0 & 0 & 0 & 0 \\ 0 & 0 & 0 & 0 & 0 & 0 \end{bmatrix}$$

**Figure 5.** Confusion matrix of water quality grades in the Longdong section.

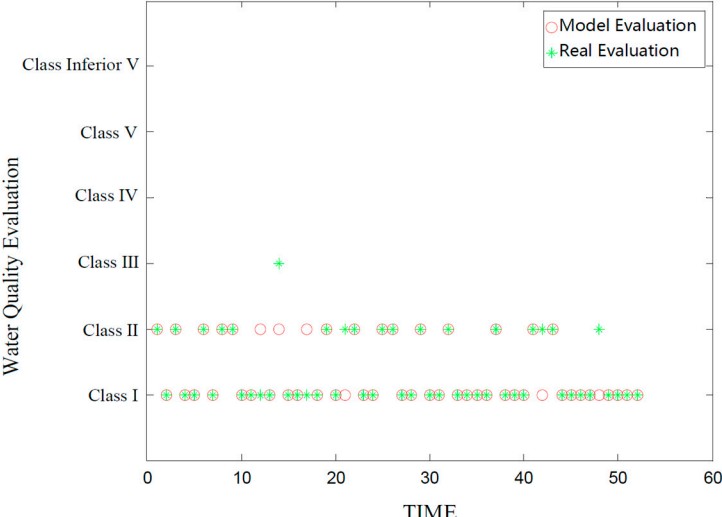

**Figure 6.** Water quality evaluation of Longdong section.

According to the confusion matrix, the water quality grade was correctly judged for 46 weeks and incorrectly judged for 6 weeks; the total correct judgment rate reached 88.46%. The correct judgment rates of class I and II water quality were 94.12% and 82.35%, respectively. Class I water quality was misjudged as class II for two weeks, class II was misjudged as class I for three weeks and class III was misjudged as class II for one week.

As shown in Figure 6, the water quality was mainly classified as class I and class II for the 52-week investigation period. In the 12th and 17th weeks, water quality in class I was incorrectly classified as class II; in the 21st, 42nd and 48th weeks, the water quality in class II was incorrectly classified as class I; and in the 14th week, the water quality in class III was incorrectly classified as class II. Results are shown in Table 4:

**Table 4.** Misjudgment of water quality grade in Longdong section.

| Week | True Water Quality | Misjudged Quality |
| --- | --- | --- |
| 12 | I | II |
| 14 | III | II |
| 17 | I | II |
| 21 | II | I |
| 42 | II | I |
| 48 | II | I |

The water quality evaluation results presented above indicate that an ideal state was achieved in all investigated river basins. In the Yellow River in Lanzhou Xincheng Bridge section and Panzhihua, Sichuan province, in the Yangtze River Basin Longdong section, the water levels were correctly classified at rates of 90.38% and 88.46%, respectively, indicating that the trained T-S fuzzy neural network achieved satisfactory quality evaluation with good generalization ability.

### 4.2. Water Quality Prediction Model

4.2.1. Water Quality Prediction of Lanzhou Xincheng Bridge Section

For prediction with the ARIMA model, we used E-views10 software, which is created by IHS Global Inc in the State of California, United States. An original sequence diagram was generated to intuitively judge whether the data sequence was stable, as shown in Figure 7.

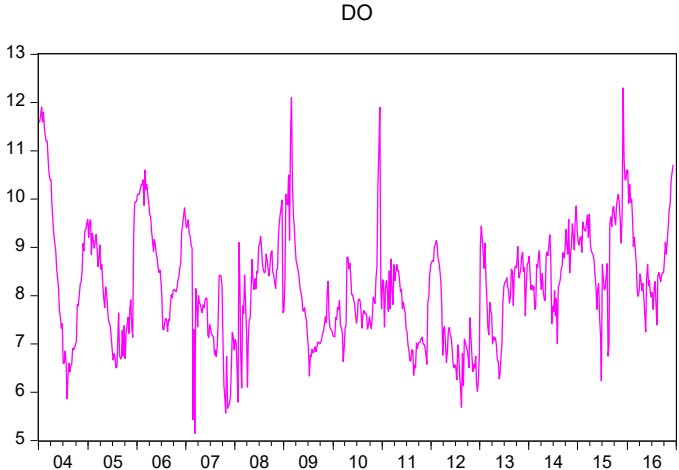

**Figure 7.** Original dissolved oxygen values.

In Figure 7, the *x*-axis shows the time in months, and the *y*-axis shows dissolved oxygen in milligrams per liter. The series in the figure does not indicate an obvious trend or seasonality, so it was preliminarily judged to be stationary series [35]. For further confirmation, an ADF unit root test was performed. The results show that the ADF test statistic is less than the critical value at levels of 1%, 5% and 10%, so the dissolved oxygen series is stationary and does not require differential processing [36]. An autocorrelation diagram and partial autocorrelation diagram were generated for model order determination, as shown in Figure 8.

As shown in Figure 8, the trailing and censoring characteristics of the autocorrelation function and partial autocorrelation function are very obvious in the dissolved oxygen data series. The decay of the autocorrelation function is very slow, which is a typical characteristic of trailing. However, the partial autocorrelation function rapidly decays to within two times the standard deviation after two steps, so the AR(2) model is determined. The estimation results of the model are shown in Table 5.

The $R^2$ of the model reached 79.34%, and an adaptability test was conducted on the AR(2) model was conducted, i.e., a white noise test of the residuals, as shown in Figure 9 [37]:

As shown in Figure 9, the autocorrelation and partial autocorrelation functions of the residual sequence both fall within two standard deviations, and the P value is significantly greater than 0.05. Therefore, it is a white noise sequence, indicating that the useful information in the original sequence has been extracted and the model has passed the adaptability test. Then, the AR(2) model established in this paper is used to predict the test data. A comparison between the predicted value and the real value is shown in Figure 10.

**Figure 8.** Autocorrelation and partial autocorrelation of dissolved oxygen.

**Table 5.** Parameter estimation of the AR(2) model.

| Variable | Coefficient | Standard Error | T Statistic | *p* Value |
|---|---|---|---|---|
| C | 8.16 | 0.20 | 40.13 | <0.001 |
| AR(1) | 0.78 | 0.04 | 20.46 | <0.001 |
| AR(2) | 0.11 | 0.04 | 2.99 | <0.001 |

**Figure 9.** White noise test of residual value.

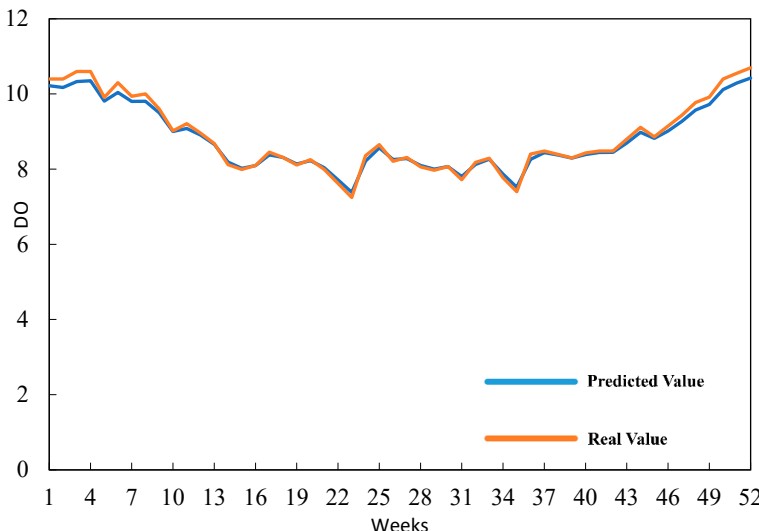

**Figure 10.** ARIMA Prediction Diagram.

In Figure 10, the *x*-axis represents time, the *y*-axis represents the concentration of DO, the blue line represents the predicted value of the ARIMA model and the orange line represents the real value. We can intuitively observe that the coincidence degree of the two lines is relatively high, which indicates that the prediction effect of the ARIMA model is strong. Before using wavelet neural network for prediction, the irrelevant noise in the data is removed. The empirical mode decomposition (EMD) of the dissolved oxygen sequence was carried out by MATLAB, as shown in Figure 11:

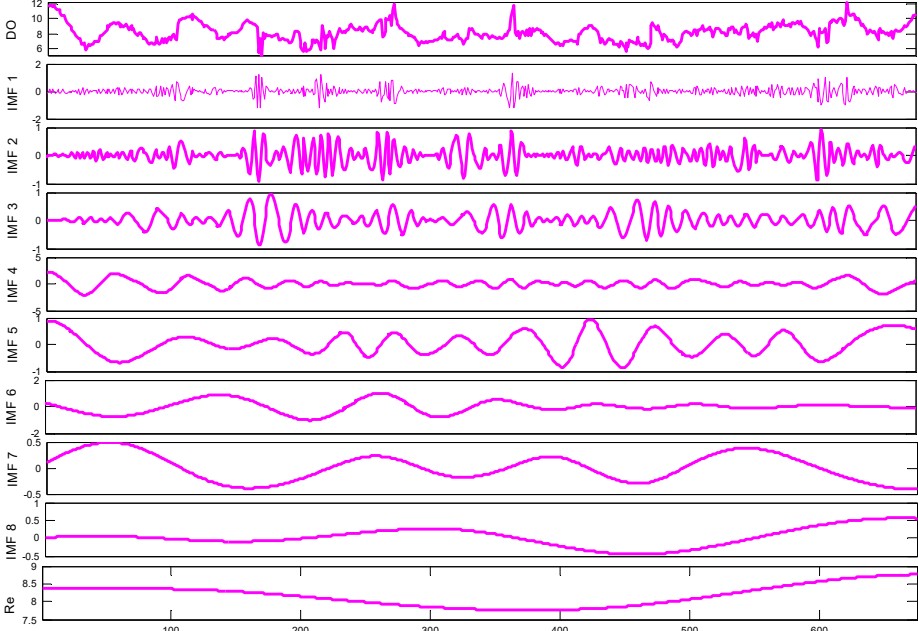

**Figure 11.** EMD diagram of dissolved oxygen.

The original data series were decomposed into eight intrinsic mode functions (IMFs) and a residual sequence. The frequency decreases from IMF1 to IMF8, and each IMF has its own unique frequency and cycle [38]. Due to the high-frequency property of IMF1 and its chaotic fluctuation trend, it was removed on the basis of the original data sequence too obtain a new sequence after denoising. The original sequence was denoised as shown in Figure 12:

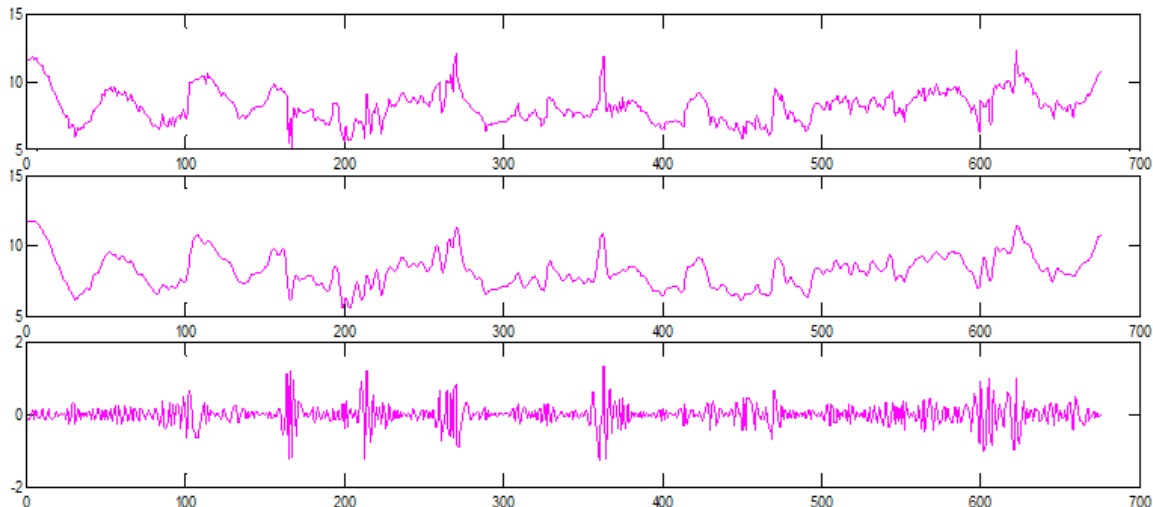

**Figure 12.** Comparison of dissolved oxygen sequence before and after denoising.

The denoised sequence is smoother than the original sequence, showing its original fluctuation trend more clearly. Therefore, the denoised sequence can be used for subsequent experimental demonstration and analysis.

First, a three-layer forward neural network is created, the parameters of each network are initialized and a training set is constructed to train the wavelet neural network. The data for weeks T-1, T-2, T-3 and T-4 are used to predict the number of neurons in week T. Therefore, the number of neurons in the input layer and output layer is four and one, respectively, and the number of neurons in the hidden layer is nine, as determined by the trial-and-error method; therefore, the structure of the wavelet neural network proposed in this paper is 4–9–1. The Morlet wavelet function is selected as the wavelet basis function, the number of iterations is set as 100, the learning probability is 0.001 and the training target is $10^{-6}$. The weight and parameters of the network are modified by gradient correction method so that the predicted output is close to the desired output [39]. The predicted result is output and compared with the real value, as shown in Figure 13.

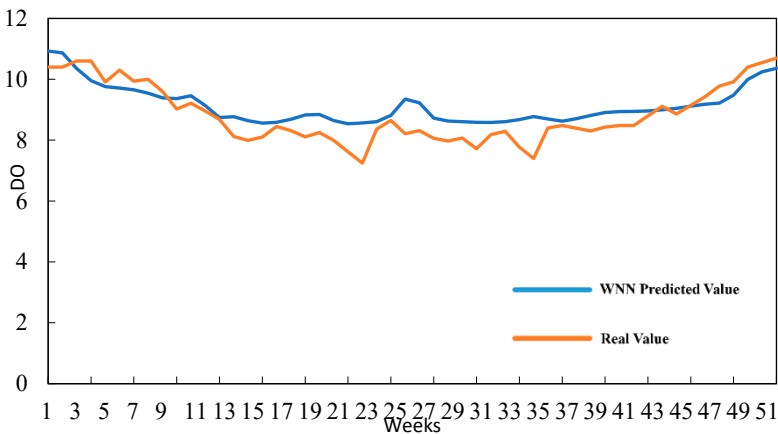

**Figure 13.** WNN prediction diagram.

The bat algorithm is used to determine the coefficients of each model, and the process of determining the optimal weight is transformed into the process of hunting prey by bats; therefore, the combined prediction model of dissolved oxygen in Lanzhou Xincheng Bridge section of the Yellow River Basin is as follows:

$$P_{DO} = 0.1495P_{WNN} + 0.8505P_{ARIMA}$$

The combined prediction model of permanganate and ammonia nitrogen was obtained according to the same steps:

$$P_{COD_{Mn}} = 0.7852 P_{WNN} + 0.2148 P_{ARIMA}$$

$$P_{NH_3-N} = 0.3274 P_{WNN} + 0.6726 P_{ARIMA}$$

4.2.2. Forecast Results and Analysis

The MAPE, MAE, RMSE of DO, COD$_{Mn}$ and NH$_3$-N in 2016 under the single model and combined model prediction, as well as the degree of improvement of the combined model compared with the single model prediction ability were obtained according to the modeling process described above, as shown in Tables 6 and 7, respectively.

**Table 6.** Comparison of water quality index predictions.

| Water Quality Index | Prediction Accuracy Index | ARIMA | WNN | Combined Model |
|---|---|---|---|---|
| | MAPE | 2.73% | 5.57% | 2.58% |
| DO | MAE | 0.2395 | 0.4716 | 0.2248 |
| | RMSE | 0.2867 | 0.5587 | 0.2758 |
| | MAPE | 19.64% | 12.97% | 11.96% |
| COD$_{Mn}$ | MAE | 0.4885 | 0.3517 | 0.3160 |
| | RMSE | 0.6853 | 0.4611 | 0.4079 |
| | MAPE | 16.18% | 17.40% | 13.85% |
| NH$_3$-N | MAE | 0.0320 | 0.0307 | 0.0268 |
| | RMSE | 0.0427 | 0.0324 | 0.0324 |

**Table 7.** Improvement in the predictive ability of the combined model relative to the single model.

| | DO | COD$_{Mn}$ | NH$_3$-N |
|---|---|---|---|
| AI$_{ARIMA}$ | 5.49% | 39.10% | 14.40% |
| AI$_{WNN}$ | 53.68% | 7.79% | 20.40% |

For the three water quality indicators, the MAPE, MAE and RMSE values of the combined model are lower than those of the single model, and the prediction effect is better. In the prediction of DO, COD$_{Mn}$ and NH$_3$-N, the prediction accuracy of the combined model is improved by 5.49% and 53.68%, 39.10% and 7.79%, and 14.40% and 20.40% compared with the ARIMA model and WNN, respectively.

For a particular set of data, a higher weight is assigned to a single model of the combined model, which indicates that the method has a better predictive ability. Taking DO as an example, the prediction accuracy of the ARIMA model is higher than that of WNN, and it has a better prediction ability. In the ARIMA-WNN model, the bat algorithm was used to calculate the weights of the two individual models as 0.8505 and 0.1495, respectively, meaning that the individual model with better prediction ability had more weight and proving the reliability of to the bat algorithm to determine the weight coefficient in the combined model. The combined model of water quality prediction proposed in this paper was compared with the prediction results of each individual model, the BP neural network and LSSVM. The results are shown in Figures 14–17:

As shown in the figures presented above, the MAPE, MAE and RMSE of the combined model are significantly lower than those of the comparison models. The fitting figures of the real and predicted values also show that the water quality prediction model proposed in this paper has a higher fitting degree to the real data.

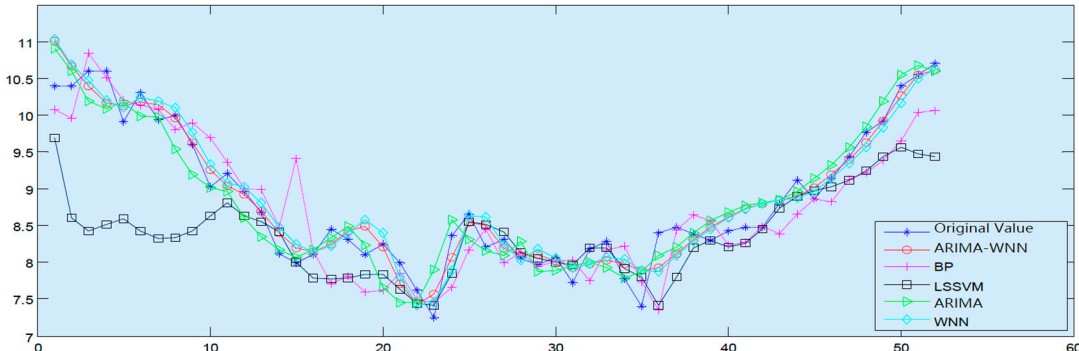

**Figure 14.** Comparison of each model fitting for dissolved oxygen.

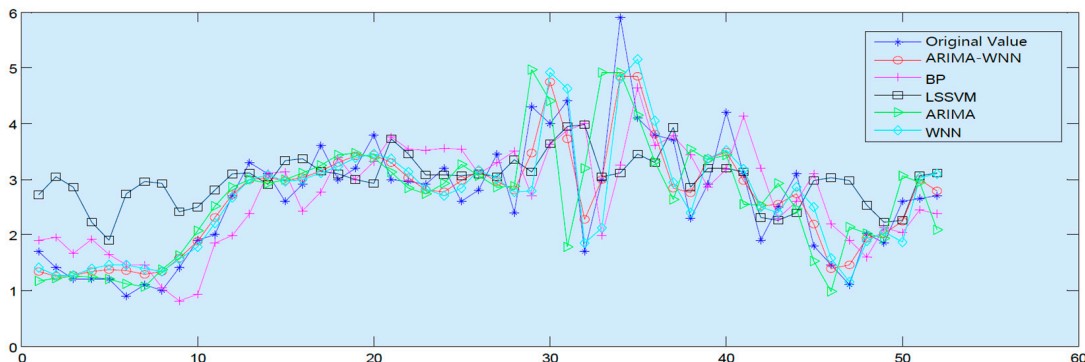

**Figure 15.** Comparison of each model fitting for CODMn.

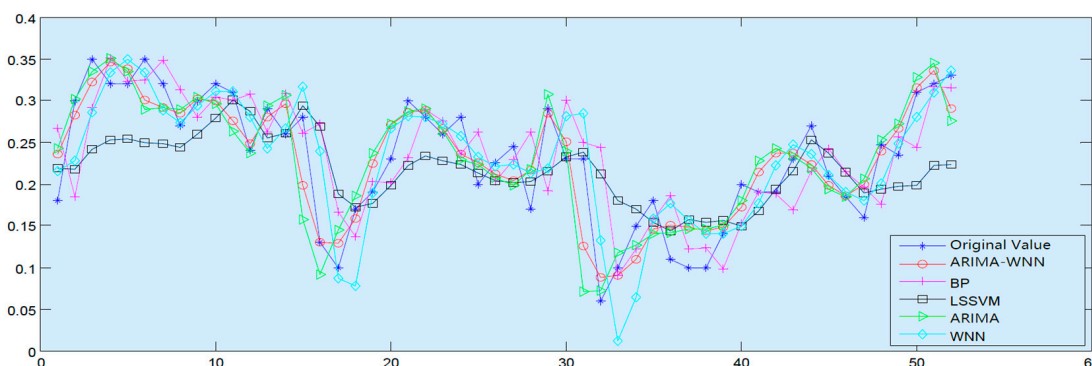

**Figure 16.** Comparison of each model fitting for NH$_3$-N.

4.2.3. Water Quality Prediction of Longdong Section

In order to verify the predictive ability of the combined model proposed in this paper for different water systems, the water quality index data of the Longdong section of Panzhihua in Sichuan Province of the Yangtze River Basin are selected for prediction and analysis. The modeling results are as follows:

$$P_{DO} = 0.1044P_{WNN} + 0.8956P_{ARIMA}$$

$$P_{COD_{Mn}} = 0.0432P_{WNN} + 0.9568P_{ARIMA}$$

$$P_{NH_3-N} = 0.8646P_{WNN} + 0.1354P_{ARIMA}$$

The improvement in the prediction accuracy and combined model prediction ability of the single and combined models for each indicator are compared in Tables 8 and 9.

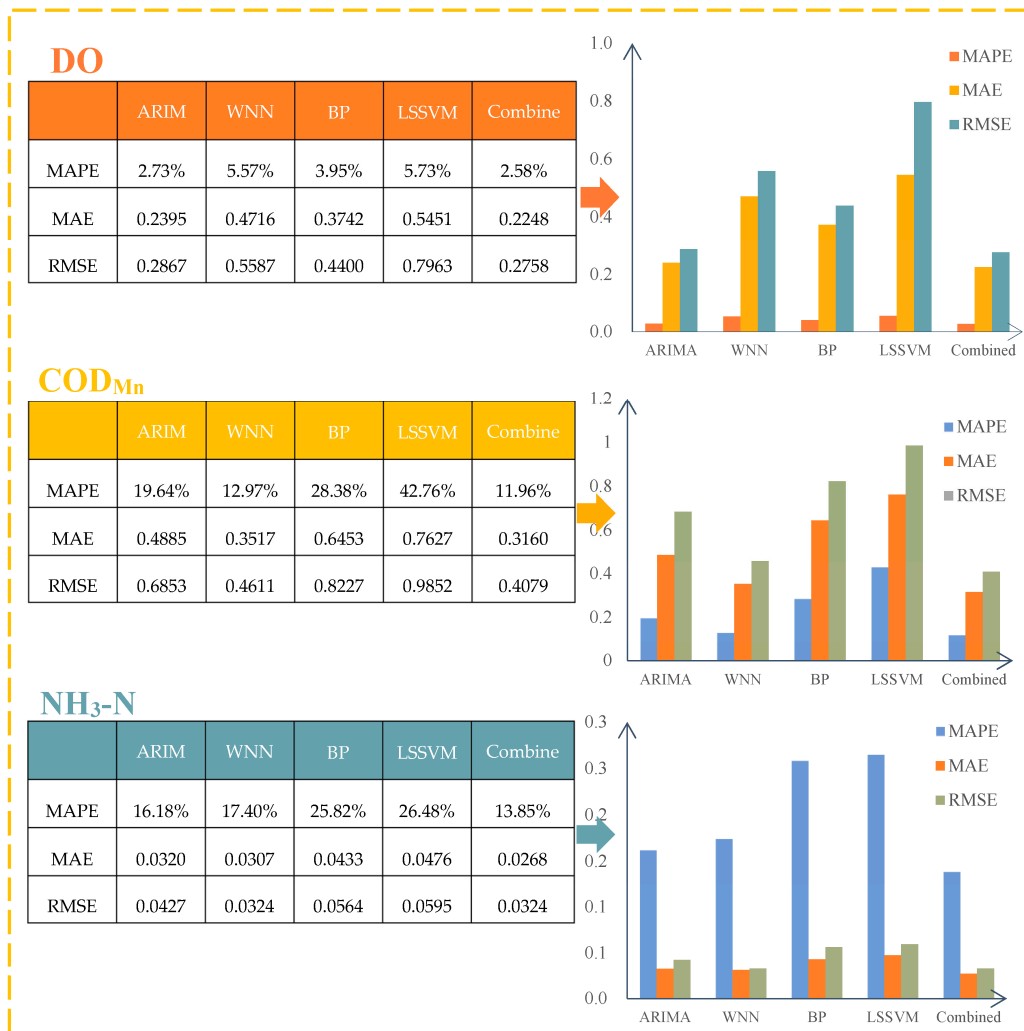

**Figure 17.** Comparison of the combined model with other models.

As shown in Table 8, the prediction effect of the combined model is better than that of each single model. For DO, COD$_{Mn}$ and NH$_3$-N, the combinatorial model outperformed the ARIMA model by 2.99%, 6.73% and 68.06%, respectively, in terms of prediction ability and outperformed the prediction of the WNN model by, 58.33%, 66.17% and 11.67%, respectively.

In the prediction of various indicators of the Longdong section of Panzhihua in Sichuan in the Yangtze River Basin, the bat algorithm still assigns more weight to the single model with better prediction ability and less weight to the single model with poor prediction ability. With respect to DO, the MAPE value predicted by the wavelet neural network is 4.68%, compared with 2.01% for the ARIMA model, which suggests that the ARIMA model achieves better predictive performed; therefore, in the combinatorial model, the bat algorithm assigns it a weight of 0.8956, whereas the WNN only assigns it a weight of 0.1044. In comparison with a BP neural network and LSSVM, the prediction accuracy of the combined model proposed in this paper is higher; the comparison results are shown in Figures 18–21.

In the prediction of various water quality indices in the Longdong section of Panzhihua, Sichuan, in the Yangtze River Basin, the prediction effect of the combined model is better and the accuracy is higher than that of the single model, the BP neural network and the LSSVM, indicating that the combined model proposed in this paper is suitable for water quality prediction in different river basins, with satisfactory generalization performance [40].

**Table 8.** Comparison of single and combined model prediction accuracy of each indicator.

| Water Quality Indicator | Prediction Accuracy Indicator | ARIMA | WNN | Combined Model |
|---|---|---|---|---|
| | MAPE | 2.01% | 4.68% | 1.95% |
| DO | MAE | 0.1860 | 0.4116 | 0.1815 |
| | RMSE | 0.2481 | 0.5201 | 0.2446 |
| | MAPE | 6.39% | 17.62% | 5.96% |
| $COD_{Mn}$ | MAE | 0.1169 | 0.3220 | 0.1045 |
| | RMSE | 0.1578 | 0.5055 | 0.1498 |
| | MAPE | 27.02% | 9.77% | 8.63% |
| $NH_3$-N | MAE | 0.0236 | 0.0092 | 0.0088 |
| | RMSE | 0.0315 | 0.0144 | 0.0148 |

**Table 9.** Improvement in the predictive power of the combined model compared to a single model.

| | DO | $COD_{Mn}$ | $NH_3$-N |
|---|---|---|---|
| $AI_{ARIMA}$ | 2.99% | 6.73% | 68.06% |
| $AI_{WNN}$ | 58.33% | 66.17% | 11.67% |

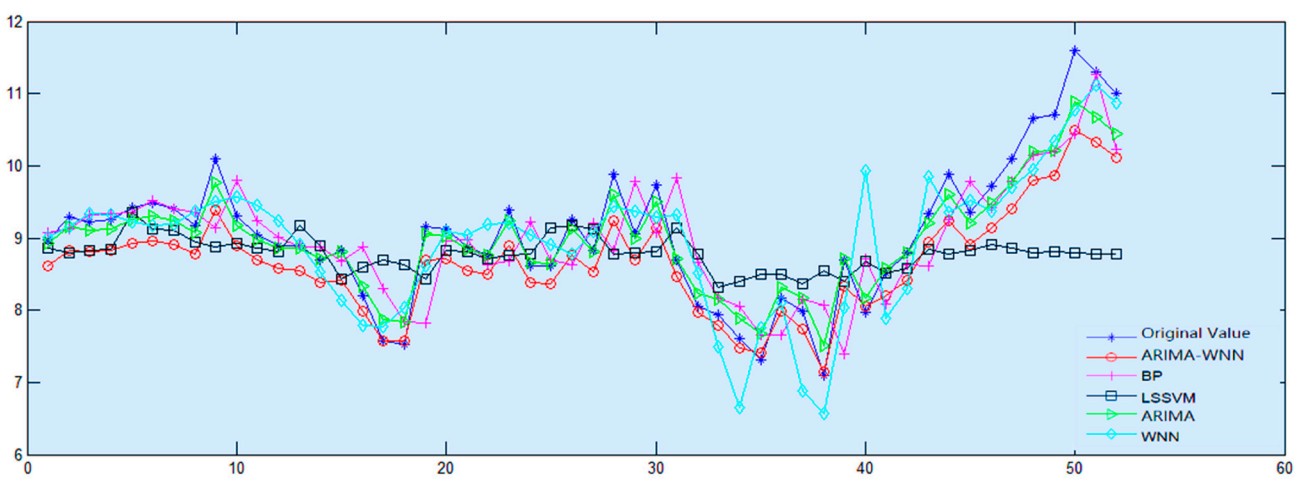

**Figure 18.** Comparison of model prediction of dissolved oxygen.

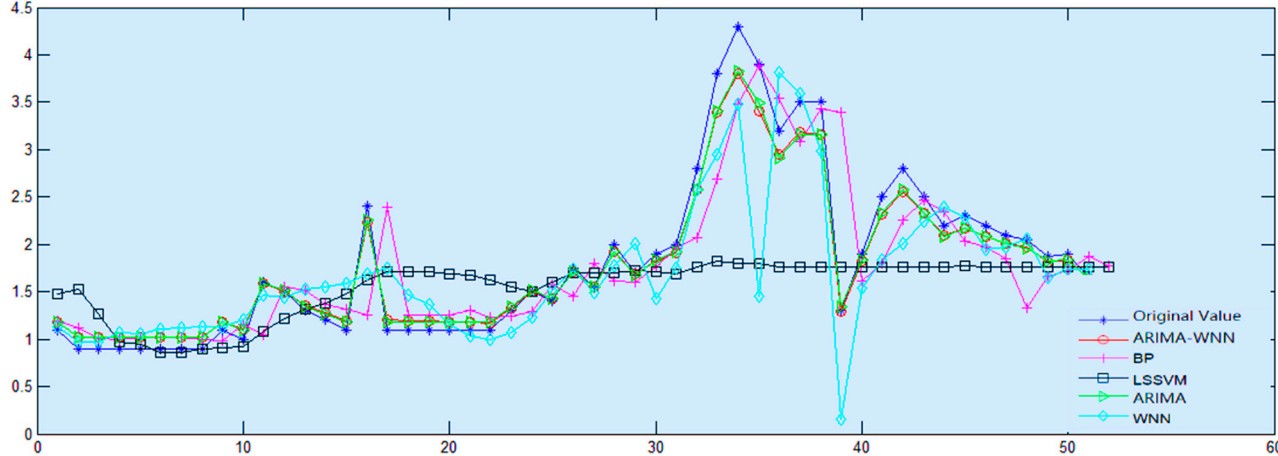

**Figure 19.** Comparison of model prediction of CODMn.

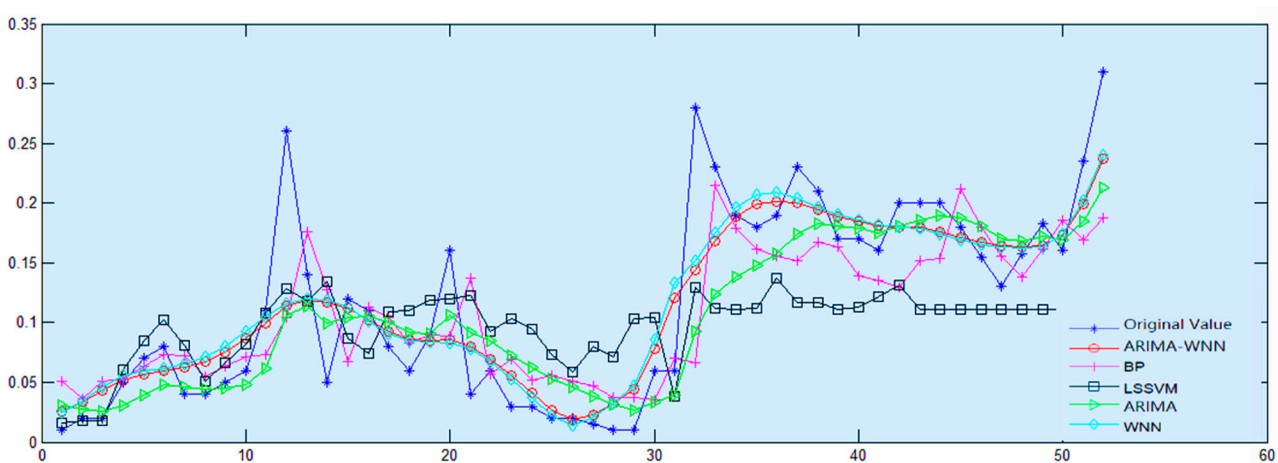

**Figure 20.** Comparison of model prediction of NH₃-N.

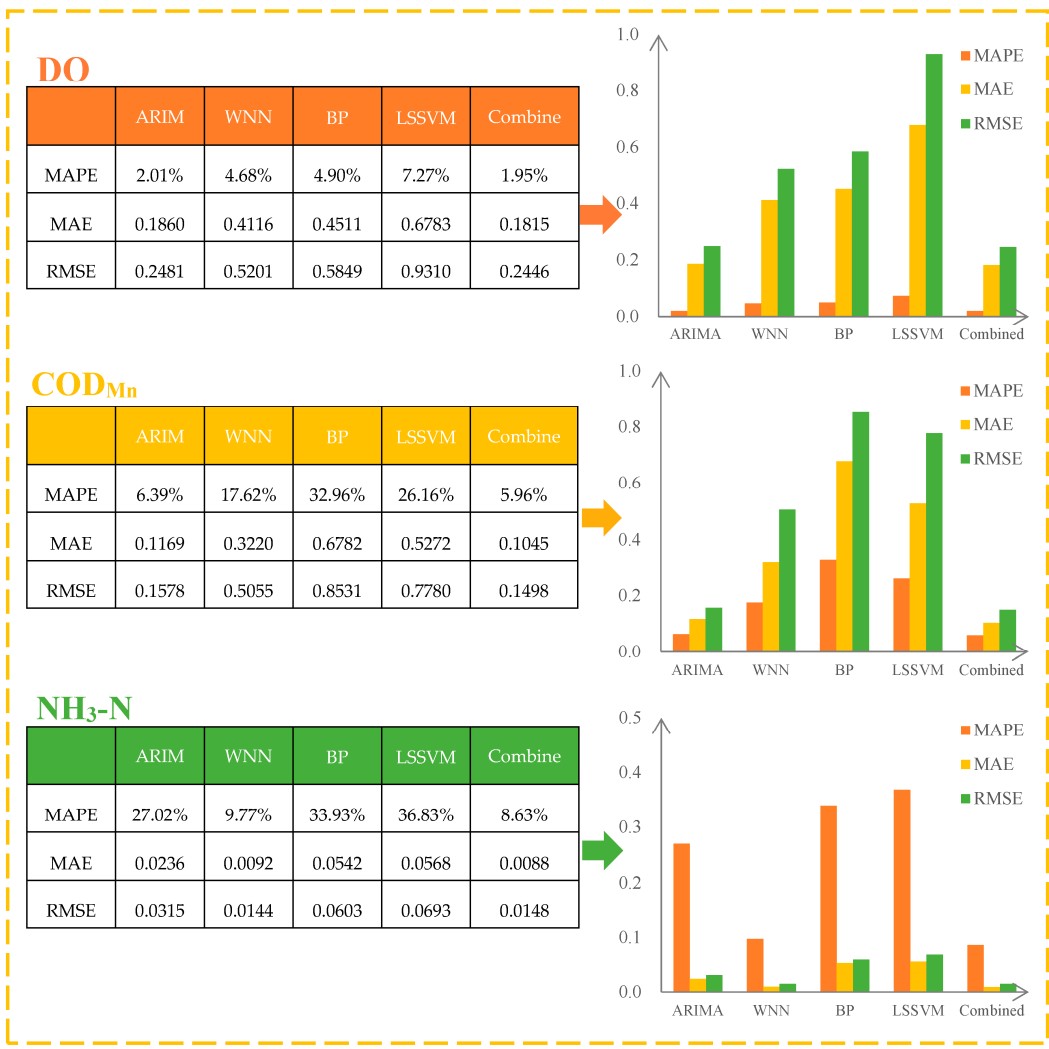

**Figure 21.** Combined model compared to other models.

### 4.2.4. Water Quality Evaluation in 2016

The results in presented in Section 4.1.3 prove that the trained T-S fuzzy neural network described in this paper has excellent ability in the determination of water quality grade. On this basis, the water quality index data of two sections predicted for 2016 were substituted for water quality evaluation. The results are shown in Tables 10 and 11 and Figures 22 and 23.

**Table 10.** Water quality evaluation of Lanzhou Xincheng Bridge Section in 2016.

| Week | Evaluation Results | True Water Quality Rating | Week | Evaluation Results | True Water Quality Rating | Week | Evaluation Results | True Water Quality Rating | Week | Evaluation Results | True Water Quality Rating |
|---|---|---|---|---|---|---|---|---|---|---|---|
| 1 | II | II | 14 | II | II | 27 | II | II | 40 | II | III |
| 2 | II | II | 15 | II | II | 28 | II | II | 41 | II | II |
| 3 | II | II | 16 | II | II | 29 | III | III | 42 | II | II |
| 4 | III | II | 17 | II | II | 30 | II | II | 43 | I | II |
| 5 | II | II | 18 | II | II | 31 | III | III | 44 | II | II |
| 6 | II | II | 19 | II | II | 32 | I | I | 45 | II | II |
| 7 | II | II | 20 | II | II | 33 | II | II | 46 | II | II |
| 8 | II | II | 21 | II | II | 34 | III | III | 47 | II | II |
| 9 | II | II | 22 | III | II | 35 | II | III | 48 | II | II |
| 10 | II | II | 23 | II | II | 36 | II | II | 49 | III | II |
| 11 | II | II | 24 | II | II | 37 | II | II | 50 | II | II |
| 12 | II | II | 25 | II | II | 38 | II | II | 51 | II | II |
| 13 | I | II | 26 | II | II | 39 | II | II | 52 | II | II |

**Table 11.** Water quality evaluation of Longdong section in 2016.

| Week | Evaluation Results | True Water Quality Rating | Week | Evaluation Results | True Water Quality Rating | Week | Evaluation Results | True Water Quality Rating | Week | Evaluation Results | True Water Quality Rating |
|---|---|---|---|---|---|---|---|---|---|---|---|
| 1 | I | I | 14 | I | I | 27 | I | I | 40 | II | II |
| 2 | I | I | 15 | I | I | 28 | I | I | 41 | II | II |
| 3 | I | I | 16 | I | I | 29 | I | I | 42 | II | II |
| 4 | II | I | 17 | I | I | 30 | I | I | 43 | II | II |
| 5 | I | I | 18 | II | I | 31 | II | I | 44 | II | II |
| 6 | I | I | 19 | I | I | 32 | II | II | 45 | II | II |
| 7 | I | I | 20 | I | II | 33 | II | II | 46 | I | II |
| 8 | I | I | 21 | I | I | 34 | III | III | 47 | II | II |
| 9 | I | I | 22 | I | I | 35 | II | II | 48 | II | II |
| 10 | I | I | 23 | I | I | 36 | II | II | 49 | I | II |
| 11 | I | I | 24 | I | I | 37 | II | II | 50 | II | II |
| 12 | I | II | 25 | I | I | 38 | II | II | 51 | I | II |
| 13 | I | I | 26 | II | I | 39 | I | II | 52 | II | II |

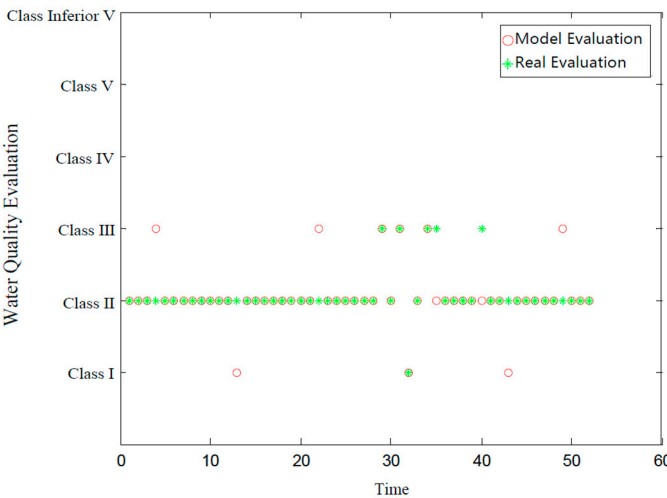

**Figure 22.** Water quality evaluation of Lanzhou Xincheng Bridge section in 2016.

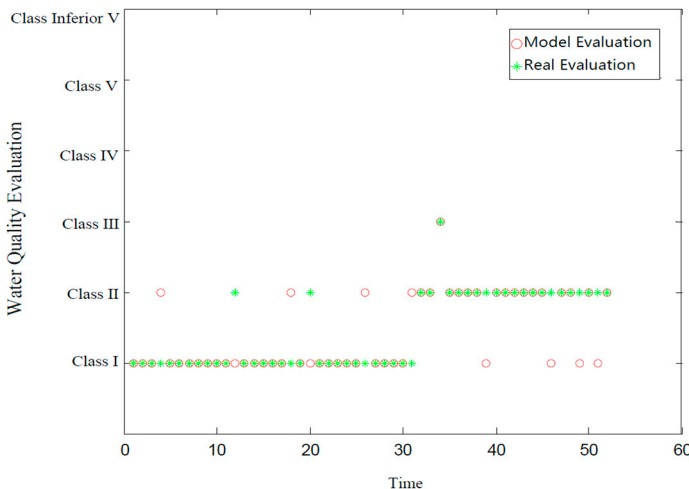

**Figure 23.** Water quality evaluation of Longdong section in 2016.

Water quality evaluation using the T-S fuzzy neural network trained as described in this paper revealed that among the 52 weeks of 2016, water quality was misjudged in the Lanzhou Xincheng Bridge section of the Yellow River Basin and the Longdong section of Panzhihua, Sichuan, the Yangtze River Basin for 7 and 10 weeks, respectively, with total correct judgment rates of 86.54% and 80.77%, respectively. Because the input data contain errors, this result is acceptable, verifying the reliability of the water quality evaluation model trained as described in this paper.

## 5. Conclusions

Water quality evaluation and prediction are not two completely independent procedures. On the contrary, they form a mutually dependent system and complement each other. Accordingly, in this study, we established a water quality evaluation–prediction system, established a water quality evaluation model using a T-S fuzzy neural network and constructed research samples by interpolating water quality index data evenly distributed on the basis of each index grading standard stipulated in the *Environmental Quality Standards for Surface Water*. A stratified sampling method is used to construct training samples. The trained T-S fuzzy neural network was applied to the water quality evaluation of the Lanzhou Xincheng Bridge section in the Yellow River Basin and the Longdong section in the Sichuan Panzhihua section in the Yangtze River Basin, with total positive water quality grade evaluation rates of 90.38% and 88.46%, respectively, indicating the positive water quality evaluation effect and generalizability of the model.

For the prediction of water quality, in this paper, we proposed a new combined model, ARIMA-WNN, which establishes the combined prediction model for each water quality index of the two basins and compares the prediction results with the combined model. The results show that compared with the single model, the combined model (ARIMA-WNN) has a higher prediction accuracy, and the prediction ability can be improved by up to 68.06%. Compared with commonly used water quality prediction models (BP neural network and LSSVM), we found that the MAPE, MAE and RMSE of the combined model are significantly lower, which demonstrates the excellent water quality prediction ability of the combined model.

Determining reasonable weight coefficients for each single model in the combined model is the basis for obtaining accurate prediction results, and in this study, we used the bat algorithm to achieve this process. Swarm intelligence optimization algorithms have developed rapidly in recent years, and a variety of new methods have emerged in succession [41]. Determining the optimal weight is a subject that can be studied in depth, and additional methods should be proposed and tested in subsequent work [42].

**Author Contributions:** Conceptualization, G.J.; Methodology, G.J.; Writing—original draft, F.W. (Fei Wang), F.W. (Fanjuan Wang), H.L., F.Z., J.C. and J.J.; Writing—review & editing, S.C.; Supervision, G.J. and Z.W. All authors have read and agreed to the published version of the manuscript.

**Funding:** The research was supported by National Natural Science Foundation of China Under Grant No.41271038.

**Institutional Review Board Statement:** Not applicable.

**Informed Consent Statement:** Not applicable.

**Data Availability Statement:** The data presented in the present study are available on request from the corresponding author.

**Conflicts of Interest:** The authors declare no conflict of interest.

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
