# Peer review of "Water Quality Evaluation and Prediction Based on a Combined Model"

_applsci, doi:10.3390/app13031286_

Round 1

Reviewer 1 Report

I find this work to be very interesting and I believe it will interest the general audience of this journal because authors have been able to used a combined model (ARIMA-WNN) to predict and evaluate water quality. They were able to compare the performance of the different models and the combined model using evaluation metrics like MAPE, RMSE, MAE and AI. I have the following comments:

- Figures 4,6,16,18,19,20,22,23 have a very low resolution. These figures should be re-presented.

- Why is p-value equal to zero in Table 5? What does that mean?

- Why didn't authors approach there prediction by having a train, validation (you can use something like cross validation or bootsrapping) and test model? This approach will make the analysis and the result to be better.

Author Response

Dear Reviewer:

Thank you for your comments concerning our manuscript entitled ‘Water Quality Evaluation and Prediction Based on Combined Model’ (applsci-2121413). Your comments are all valuable and very helpful for revising and improving our paper, as well as the important guiding significance to our researches. We have studied comments carefully and have made correction which we hope meet with approval. We mark all the changes in red in the revised manuscript.

The following is a point-to-point response to your comments.

Minor points

Comment 1: Figures 4,6,16,18,19,20,22,23 have a very low resolution. These figures should be re-presented.

Response: Thank you for your careful work and point out the issues of low figure resolution. According to your comments, we have represented figures 4,6,16,18,19,20,22,23, which are given at their first appearance in line 315, line 337, line 450, line 501, line 503, line 505, line 542 and line 545 in red. In addition, we have carefully rechecked the resolution of all figures.

Comment 2: Why is p-value equal to zero in Table 5? What does that mean? 

Response: Thank you for your comments. The data is displayed as 0, not that the data is really zero, but a very small number that has all zeros before the reserved significant decimal point, so the fixed display is 0.00, which is just a way to display data. The probability P value obtained by the statistical test is less than the significance level α, which means that it is very convenient to make the test inference. Thank you for pointing out such a problem, the expression in the article is indeed not rigorous, we have revised it to p<0.001 in Table 5 in line 371 in red.

Comment 3: Why didn't authors approach there prediction by having a train, validation (you can use something like cross validation or bootsrapping) and test model? This approach will make the analysis and the result to be better.

Response: Thank you for your valuable and thoughtful comments. In the process of model training, cross validation and bootsrapping are indeed effective in finding model hyperparameters, but the cost of such methods is a large amount of calculation. The data set used in this paper is not large, but the model used involves a lot of neural network parameters. The method you suggested is objectively a better approach in model training, however, in the case of limited computing resources and sufficient prior knowledge, cross validation is not selected in this paper.

Finally, we appreciate for your warm work earnestly, and hope that the correction will meet with approval.

Once again, thank you very much for your comments and suggestions.

Best wishes.

Corresponding Authors: Guimei Jiao *, Shaokang Chen, Fei Wang, Zhaoyang Wang, Fanjuan Wang, Hao Li, Fangjie Zhang, Jiali Cai, Jing Jin

E-mails: [email protected], [email protected], [email protected],

[email protected], [email protected], [email protected],

[email protected], [email protected], [email protected]

Reviewer 2 Report

The paper is interesting but requires some minor corrections.

1. ARIMA, WNN, BP, LSSVM should be in full words in Introduction.

2. On page 8, line 276, I suggest "Chemical Oxygen Demand permanganate" instead of "permanganate".

3. In figures 4 and 6, what is the unit of time?

4. In figure 7, what is in x-axis and y-axis?

5. On page 12, line 358, which figure?

6. On page 13, line 366, which figure?

7. In table 5, what is parameter?

8. There is no discussion for figure 10.

9. On page 18, why figure 1?

10. On page 21, why figure 2?

11. Some recent result can be cited. i.e. Results in Engineering 8, 100189

Author Response

Dear Reviewer:

Thank you for your comments concerning our manuscript entitled ‘Water Quality Evaluation and Prediction Based on Combined Model’ (applsci-2121413). Your comments are all valuable and very helpful for revising and improving our paper, as well as the important guiding significance to our researches. We have studied comments carefully and have made correction which we hope meet with approval. We mark all the changes in red in the revised manuscript.

The following is a point-to-point response to your comments.

Minor points

Comment 1: ARIMA, WNN, BP, LSSVM should be in full words in Introduction.

Response: Thank you for your careful work and point out the issues of intensive use of abbreviations. According to your comments, the full names of ARIMA, WNN, BP, and LSSVM are given at their first appearance in line 80, line 81, line 91 and line 92 in red, respectively. In addition, we have carefully rechecked all abbreviations in Introduction.

Comment 2: On page 8, line 276, I suggest "Chemical Oxygen Demand permanganate" instead of "permanganate". 

Response: Thank you for your detailed suggestion. We have changed "permanganate" to "Chemical Oxygen Demand permanganate" in line 277 in red.

Comment 3: In figures 4 and 6, what is the unit of time?

Response: Thank you for your detailed comment. The unit of time is week in figure 4 and 6, which we have added an explanation in line 317 and line 341 in red.

Comment 4: In figure 7, what is in x-axis and y-axis?

Response: Thank you for your detailed suggestion. Figure 7 shows the dissolved oxygen original value. The x-axis shows time in months, from 04 to 16 representing a full year, and the y-axis shows dissolved oxygen in milligrams per liter, which we have added an explanation in line 360 in red.

Comment 5: On page 12, line 358, which figure?

Response: Thank you for your detailed advice. The Figure 7 is in line 358 on page 12, which shows the dissolved oxygen original value.

Comment 6: On page 13, line 366, which figure?

Response: Thank you for your detailed suggestion. The Figure 8 is in line 366 on page 13, which shows the autocorrelation and partial autocorrelation of dissolved oxygen.

Comment 7: In table 5, what is parameter?

Response: Thank you for your thoughtful comment. I am sorry that our translation of the term is not rigorous. We have changed " Parameter" to "Coefficient" in Table 5 in line 375 in red which refers to the coefficients of variables C, AR(1) and AR(2) in the AR(2) model.

Comment 8: There is no discussion for figure 10.

Response: Thank you for your valuable and thoughtful comments. we have added the discussion about Figure 10 in line 389-392 in red on page 14.

Comment 9: On page 18, why figure 1?

Response: Thank you for your detailed advice. It may be a typographical error, and we have changed “Figure 1” to “Figure 17” in line 470 in red on page 18.

Comment 10: On page 21, why figure 2?

Response: Thank you for your detailed suggestion. It also is a typographical error, and we have changed “Figure 2” to “Figure 21” in line 536 in red on page 21.

Comment 11:Some recent result can be cited. i.e. Results in Engineering 8, 100189

Response: Thank you for your valuable and thoughtful comments. Your suggestion is of great significance to the academic nature of our article. According to your comments, we have added the citations of recent results in line 33, line 43 and line271 in red.

Finally, we appreciate for your warm work earnestly, and hope that the correction will meet with approval.

Once again, thank you very much for your comments and suggestions.

Best wishes.

Corresponding Authors: Guimei Jiao *, Shaokang Chen, Fei Wang, Zhaoyang Wang, Fanjuan Wang, Hao Li, Fangjie Zhang, Jiali Cai, Jing Jin

E-mails: [email protected], [email protected], [email protected],

[email protected], [email protected], [email protected],

[email protected], [email protected], [email protected]

Reviewer 3 Report

The main contributions of this manuscript are the T-S fuzzy neural model for evaluating water quality and the ARIMA-WNN model for predicting water quality, respectively. The workload of the manuscript is worthy of recognition, but many significant problems could be solved. Specific comments are as follows:

1. The literature research section is not a statement of previous research. The logical relationship between previous research should be sorted out and evaluated appropriately. Meanwhile, the insights of previous research to this manuscript should be pointed out.

2. The manuscript has several obvious formatting errors; please correct them carefully.

3. The images have blurred pixels and confusing order; please upload higher-resolution pictures and organize the order.

4. Please state the specific reasons why the study combines ARIMA and WNN methods

5. Please list the specific indicators for evaluating water quality, such as COD, TN, TP, etc.

6. The mechanics of the data-driven model lend themselves to the disadvantage of a solidified output model. Have the authors considered this problem and tried to solve it?

7. The novelty and significance of this study are not clear, please clarify.

Author Response

Dear Reviewer:

Thank you for your comments concerning our manuscript entitled ‘Water Quality Evaluation and Prediction Based on Combined Model’ (applsci-2121413). Your comments are all valuable and very helpful for revising and improving our paper, as well as the important guiding significance to our researches. We have studied comments carefully and have made correction which we hope meet with approval. We mark all the changes in red in the revised manuscript.

The following is a point-to-point response to your comments.

Major points

Comment 1: The literature research section is not a statement of previous research. The logical relationship between previous research should be sorted out and evaluated appropriately. Meanwhile, the insights of previous research to this manuscript should be pointed out.

Response: Thank you for your careful work and point out our weak logic in the literature research section. According to your comments, we have added the logical relationship of references, in addition, we have pointed out the inspiration of previous studies to this paper in line 118-123 , line 129-131 and line 146-148 in red.

Comment 2: The manuscript has several obvious formatting errors; please correct them carefully. 

Response: Thank you for your detailed suggestion. We have fixed formatting errors in the manuscript.

Comment 3: The images have blurred pixels and confusing order; please upload higher-resolution pictures and organize the order.?

Response: Thank you for your careful work and point out the issues of low figure resolution. According to your comments, we have represented figures 4,6,16,18,19,20,22,23, which are given at their first appearance in line 315, line 337, line 450, line 501, line 503, line 505, line 542 and line 545 in red. In addition, we have carefully rechecked the resolution of all figures.

Comment 4: Please state the specific reasons why the study combines ARIMA and WNN methods.

Response: Thank you for your detailed suggestion. We added the explanation why we combine ARIMA and WNN methods in line 193-197 in red.

Comment 5: Please list the specific indicators for evaluating water quality, such as COD, TN, TP, etc.

Response: Thank you for your detailed advice. We have listed the specific indicators for evaluating water quality , such as DO, CODMn and NH3-N in line 288-289 in red. We did not use TN, because we believe that NH3-N contains more information than TN, so it is more important and more revealing. Besides, we did not use TP, because the research content of this paper does not focus on TP. Among the three indexes listed in the manuscript, besides NH3-N, CODMn is an important organic pollution parameter that can be measured quickly, and DO is an index to measure the self-purification capacity of waterbody. Therefore, we believe that the indicators selected in this paper are highly representative and effective for water quality evaluation.

Comment 6: The mechanics of the data-driven model lend themselves to the disadvantage of a solidified output model. Have the authors considered this problem and tried to solve it?

Response: Thank you for your detailed suggestion. In our opinion, for statistical research, using data-driven model to solve problems is a common and efficient way. The question you raised really deserves our reflection, and we have considered this problem, but it is not for us to solve it now. Put it another way, we haven't found a better way to make full use of the data information while avoiding the shortcomings of the solidified output model in the overall problem-solving process.

Comment 7: The novelty and significance of this study are not clear, please clarify.

Response: Thank you for your thoughtful comment. As for the water quality evaluation, the manuscript constructs training samples by itself after interpolating uniformly distributed water quality index data, so as to improve the generalization ability of the water quality evaluation model. As for water quality prediction, a new combined model ARIMA-WNN is constructed in this paper. Since the echolocation characteristics of bat algorithm make it have better searching ability, the bat algorithm is applied in the weight determination of the combined model in this paper to obtain a more reasonable weight coefficient allocation, so as to improve the prediction accuracy of the water quality prediction model.

Finally, we appreciate for your warm work earnestly, and hope that the correction will meet with approval.

Once again, thank you very much for your comments and suggestions.

Best wishes.

Corresponding Authors: Guimei Jiao *, Shaokang Chen, Fei Wang, Zhaoyang Wang, Fanjuan Wang, Hao Li, Fangjie Zhang, Jiali Cai, Jing Jin

E-mails: [email protected], [email protected], [email protected],

[email protected], [email protected], [email protected],

[email protected], [email protected], [email protected]
